# A Framework to Understand the Progression of Cardiovascular Disease for Type 2 Diabetes Mellitus Patients Using a Network Approach

**DOI:** 10.3390/ijerph17020596

**Published:** 2020-01-16

**Authors:** Md Ekramul Hossain, Shahadat Uddin, Arif Khan, Mohammad Ali Moni

**Affiliations:** 1Complex Systems Research Group, Faculty of Engineering, The University of Sydney, Darlington, NSW 2008, Australia; arif.khan@sydney.edu.au; 2School of Medical Sciences, Faculty of Medicine and Health, The University of Sydney, Camperdown, NSW 2006, Australia; mohammad.moni@sydney.edu.au

**Keywords:** chronic disease, comorbidity, administrative data, graph theory, social network analysis

## Abstract

The prevalence of chronic disease comorbidity has increased worldwide. Comorbidity—i.e., the presence of multiple chronic diseases—is associated with adverse health outcomes in terms of mobility and quality of life as well as financial burden. Understanding the progression of comorbidities can provide valuable insights towards the prevention and better management of chronic diseases. Administrative data can be used in this regard as they contain semantic information on patients’ health conditions. Most studies in this field are focused on understanding the progression of one chronic disease rather than multiple diseases. This study aims to understand the progression of two chronic diseases in the Australian health context. It specifically focuses on the comorbidity progression of cardiovascular disease (CVD) in patients with type 2 diabetes mellitus (T2DM), as the prevalence of these chronic diseases in Australians is high. A research framework is proposed to understand and represent the progression of CVD in patients with T2DM using graph theory and social network analysis techniques. Two study cohorts (i.e., patients with both T2DM and CVD and patients with only T2DM) were selected from an administrative dataset obtained from an Australian health insurance company. Two baseline disease networks were constructed from these two selected cohorts. A final disease network from two baseline disease networks was then generated by weight adjustments in a normalized way. The prevalence of renal failure, fluid and electrolyte disorders, hypertension and obesity was significantly higher in patients with both CVD and T2DM than patients with only T2DM. This showed that these chronic diseases occurred frequently during the progression of CVD in patients with T2DM. The proposed network-based model may potentially help the healthcare provider to understand high-risk diseases and the progression patterns between the recurrence of T2DM and CVD. Also, the framework could be useful for stakeholders including governments and private health insurers to adopt appropriate preventive health management programs for patients at a high risk of developing multiple chronic diseases.

## 1. Introduction and Background

Type 2 diabetes mellitus (T2DM) is a chronic disease that occurs when the body becomes resistant to insulin and/or cannot make enough insulin in the pancreas [1]. Patients with T2DM are at greater risk of developing cardiovascular disease (CVD). CVD, which includes congestive heart failure (CHF), cardiac arrhythmias, valvular disease, pulmonary circulation disorders, and peripheral vascular disorders, is one of the leading causes of death in people with T2DM in most countries worldwide and can account for 50% or more of deaths due to T2DM [2]. In 2011, CVD, diabetes and chronic kidney disease (CKD) were the major cause of death in Australia, accounting for 14% of all deaths, where about 7% of all deaths were due to CVD and diabetes together [3]. In clinical literature, CVD and T2DM often occur together. Patients with T2DM have over twice the risk of occurrence of CVD than patients without T2DM [4,5]. This is partly because of various common risk factors between CVD and T2DM, such as obesity, old age, hypertension and chronic kidney disease. There are also complex relationships between CVD and T2DM, and each of them may be caused by other diseases. As a result, they are more likely to occur together in an individual. The co-occurrence of these conditions is known as comorbidity [6]. The clinical management for people with CVD and T2DM is more expensive, complex and time-consuming than for the people with a single disease. Alongside the projected increase in the prevalence of chronic diseases, the presence of CVD and T2DM as comorbidities exerts a significant social and health burden, often resulting in higher healthcare costs [7]. Although the comorbidity of chronic diseases is receiving more attention, most studies to date focus on understanding the progression of a single chronic disease, and fewer studies investigate the relationship of multiple chronic diseases [8]. Thus, the increased prevalence of comorbidity and its impact on the health of the population and the healthcare system are not clear. In particular, greater research attention is required for the chronic disease cohort with both CVD and T2DM as the data shows an increased risk of hospitalization and death for the patients with both CVD and T2DM compared to patients with only T2DM [9]. Therefore, if we can identify those diabetic patients with a risk of CVD based on their past medical data, preventive measures can be taken to increase the quality of care and reduce treatment costs. For the patients’ health information, a potential data source can be hospital admission and discharge data which contain standardized ICD (International Classification of Diseases) codes [10]. Analysis of these administrative data using data mining and social network analysis can help us to understand the progression of chronic disease comorbidities.

The development of effective disease progression modelling depends on the understanding of the disease progression pathway. In the literature, a considerable amount of work has been done in the related field of understanding the comorbidity of chronic disease progression. There are mainly three types of approaches (i.e., the statistical method, machine learning and data mining approach, and network-based approach) applied to understand the disease progression and develop the risk prediction model [11]. Rule-based scoring is a widely used statistical method to understand disease progression as well as risk prediction. It focuses on the clinical and empirical understanding of symptoms, prevalence and disease comorbidities [12,13]. In these models, scores are assigned to various physiologically observable symptoms, demographic risk factors and comorbidity conditions to assess the severity of a patient. Various rule-based scoring methods have been developed over the years to understand disease progression [12,13,14,15]. In 1987, Charlson et al. proposed the Charlson Comorbidity Index to predict the 10-year mortality for a patient by ranking a range of demographic information (e.g., age and sex) and comorbid conditions (e.g., cancer, heart disease and AIDS) [12]. The Elixhauser Index [16] shows slightly better performance for predicting mortality beyond 30 days [17,18]. Some other rule-based models such as APACHE-II (Acute Physiology and Chronic Health Evaluation-II) [13] and SAPS (Simplified Acute Physiology Score) [19] were proposed to assess intensive care unit (ICU) patients’ health conditions in the first 24 hours of admission. The results of the diagnostic tests are considered as scores that are also used to assess or make a prognosis. For example, Ewing and Clarke proposed five tests (known as Ewing’s battery test) to assess the risk of cardiovascular disease in patients with diabetes [14]. In 2008, a diabetes-specific equation was proposed to understand the disease progression and estimate the 5-year risk of cardiovascular disease in T2DM patients with the use of the A1C (i.e., glycated hemoglobin) test results [15]. Although these rule-based scoring models work well in the specific healthcare setting, they are derived from clinical and empirical observation and do not test for many population cohorts with multiple comorbidities. However, chronic diseases do not occur in isolation [20]. They often share common risk factors than can be environmental, genetic and behavioral. These risk factors have a synergistic effect [21,22] on patients’ health outcomes and thus should not be considered in isolation.

Administrative data are generated during different stages of healthcare delivery and health insurance claims. These include important information about the patient and population health, such as demographic characteristics, health behaviors, clinical diagnoses and codes for procedures, laboratory results and care utilization [23]. Recently, these data have gained popularity and are used in clinical decision-making and healthcare research, such as treatment, diagnosis, understanding disease progression and disease risk prediction [16,24,25,26]. In 2001, Nichols et al. developed a research framework to estimate the prevalence and incidence of CVD (specifically congestive heart failure) in patients with T2DM using electronic health data [27]. They also identified risk factors for diabetes-associated congestive heart failure using multiple logistic regression models. Later, they updated their study to estimate the CHF incidence rate in T2DM and identify risk factors for developing CHF in patients with T2DM over 6 years of follow-up [28]. Recently, a research methodology was developed to identify the prevalence and incidence of CVD in patients with T2DM using electronic health data [29]. The study used ICD-9 (International Classification of Diseases, Ninth Edition) codes to identify the prevalence and incidence of CVD events. Some data mining and machine learning-based methods were proposed using administrative data in different healthcare research [30,31]. For example, collaborative filtering methods were proposed to understand disease progression and predict disease risk using healthcare data [30,32]. A deep learning algorithm was used for risk prediction for multiple comorbid diseases [33]. The Bayesian network [34], a combination of graph theory and probability theory, has been used to understand the comorbidity of multiple chronic diseases [35]. A risk prediction model was developed to predict the risk of progression to chronic obstructive pulmonary disease in asthma patients using electronic health data [36]. The study used the Bayesian network to develop the proposed model.

In this study, we used a network-based approach on administrative data—i.e., hospital admission and discharge data—to understand the disease progression of CVD in patients with T2DM by considering the comorbidities. In the biomedical field, the network-based approach has been used to understand the pathogenesis of diseases using gene expression and related proteins [37]. Social network analysis (SNA) is another related approach introduced in healthcare informatics [38,39]. SNA can be defined as a set of entities, such as physicians, diseases and hospitals, with some relationships between them. More recently, administrative data have been widely used to demonstrate SNA-based approaches [40]. These approaches are used to understand relations between healthcare entities [38,41] and improve the collaboration efficiency among physicians [42]. SNA techniques were applied in administrative and electronic healthcare data of CHF patients to explore the patterns of service delivery for care coordination [43]. Khan et al. [24] proposed a framework to understand the progression of T2DM using graph theory and SNA. Their proposed framework was applied to administrative healthcare data in the Australian healthcare context. The study mainly focused on understanding a single chronic disease (e.g., T2DM) rather than the progression of multiple chronic diseases. To our knowledge, there is very little research on using a network-based approach and administrative data to understand the progression of CVD in patients with T2DM.

## 2. Methods

The following section describes the process of the disease network generation as well as the selection process of the study population and ICD code range.

### 2.1. Data Description

In Australia, the two major users of administrative health data are the federal government (i.e., the universal health care system for Australia, known as Medicare) and private health insurers [44]. The hospital admission and discharge data of patients are recorded and reported in a standard format [45] to government departments and the respective private insurer (if the patient has a private insurance policy). The administrative dataset used in this study was collected from CBHS health funds company in Australia. It contained medical records of nearly 124,000 de-identified patients who received medical services between the years 1995 and 2018. The medical records included coded information on patients, hospital admission and clinical contents, which are shown in Table 1. For each hospital admission of a patient, a set of ICD codes are recorded to indicate what medical conditions the patient had at that time. The data for the patients with T2DM and CVD were considered over the full period of the research dataset as the main aim of this study is to understand the progression of CVD in patients with T2DM. To collect a significant and valid research dataset, a systematic process of filtering and cleaning was applied to the original administrative dataset. Then, the appropriate cohorts were selected from the research dataset. Some of the data filtering criteria are (a) selecting patients with at least one admission with valid ICD codes, (b) excluding duplicate records and (c) excluding ICD codes related to physical injuries, fever and vomiting.

### 2.2. Study Population

After getting the appropriate research dataset, we focus on defining the study cohorts. In order to explore the risk of CVD for T2DM patients, two cohorts are selected in this study. We refer to the first cohort of patients who were first diagnosed with T2DM and then diagnosed with CVD as *Cohort_T2DM&CVD_*. The second cohort of patients who were diagnosed with T2DM but not diagnosed with CVD at any time point of their entire admission histories is referred to as *Cohort_T2DM_*. To select the cohort, this study looked for the presence of at least one ICD code of the underlying disease in the patient admission history. There are well-defined ICD codes for T2DM and CVD as shown in Table 2. In this study, patients with CVD were identified using ICD codes for five distinct diseases: i.e., congestive heart failure, cardiac arrhythmias, valvular disease, pulmonary circulation disorders and peripheral vascular disorders. For *Cohort_T2DM&CVD_*, we searched for patients who have ICD codes for both T2DM and CVD. The search was undertaken in such a way that it resulted in only those patients who were diagnosed with T2DM in an earlier admission and with CVD at a later admission. The patients for *Cohort_T2DM_* were selected by a search criterion in which the patients have one or more ICD codes for T2DM in their entire admission records but no ICD codes for CVD at any stage of their admission records. The criteria used to select the appropriate patient records for both *Cohort_T2DM&CVD_* and *Cohort_T2DM_* are described in Table 3.

### 2.3. ICD Code Grouping

One of the main challenges in developing disease progression models is to decide which diseases and their ICD codes should be considered. The administrative dataset used in the proposed research framework contains the disease codes (i.e., ICD codes) that are encoded in both ICD-9-AM and ICD-10-AM format. Each version has more than 20,000 unique and active ICD codes [47], and these codes are likely to be present in the dataset. Consideration of all ICD codes individually could make the analysis and network visualization task complex. For this reason, the ICD codes are grouped into comorbidities so that each node of the disease network represents one of the comorbidities. Thus, the overall number of nodes is reduced into a reasonably small number of nodes. In this context, comorbidity represents a group of diseases or health conditions that have occurred together in the same patients, such as T2DM, CVD, renal failure etc. However, clinical expertise is needed to identify the ICD codes for the corresponding comorbidities, but they are often not available. Alternately, there are several common lists of comorbidities in the literature, known as comorbidity indices, that we can adopt for the analysis, including the Charlson [12], Elixhauser [16] and Charlson/Deyo comorbidity indices [48]. These comorbidity indices are used to assess the health conditions of a patient during hospital admission. The Elixhauser comorbidity index [16] was specially developed to measure comorbidity using administrative data. The index is clinically and empirically validated and uses a rule-based scoring model [17]. This study adopted the Elixhauser index for the list of comorbidities for a particular chronic disease.

### 2.4. Definition and Creation of Graph Theory-Based Terms Used in the Proposed Model

In this study, some concepts from graph theory are used to develop the proposed framework. In the following subsection, the definition and creation procedure of graph theory-based terms are discussed.

#### 2.4.1. Individual Disease Network and Its Attributes

The individual disease network represents the health trajectory of an individual patient. The health trajectory shows the patient’s disease transition from one disease to another during subsequent admissions in the hospitals over time [24]. An individual disease network is essentially a graph where each node indicates the disease and the edge between two nodes denotes that these two nodes tend to occur sequentially. Disease information comes from the ICD codes in the administrative data; i.e., hospital admission and discharge data. In the network, the edge is directional and is chosen in such a way that a disease which occurred in an earlier admission is put as a source node and a disease from a subsequent admission is put as the target node. If there are multiple diseases in any admission, then all possible disease pairs are considered. Also, when the patient has more than one disease in the same admission, all possible disease pairs from the same admission are shown as bi-directional edges. Each individual disease network has two attributes: one is the node-level attribute, called node frequency, which refers to the number of times the diseases have occurred for the patient considering all admissions; the other is the edge-level attribute, called edge weight, that refers to the numbers of times two diseases have occurred simultaneously or in consecutive admissions.

The first two parts of Figure 1 illustrate the construction process of the individual disease network. For patient 2, there are two admission episodes (i.e., E_1_ and E_2_) in the admission history. The patient was diagnosed with two diseases (i.e., D_1_ and D_2_) at episode E_1_ and one disease (i.e., D_2_) at episode E_2_. In the individual disease network, the disease D_2_ has a node frequency of 2 since it has appeared twice in the admission episode. The edge from D_1_ and D_2_ has a weight of 2 since it has appeared once in subsequent admission episodes (i.e., E_1_ and E_2_) and once in the same admission episode (i.e., E_1_). The notation for one appearance in subsequent admission episodes is as follows: D_1_ appeared in E_1_ and D_2_ appeared in E_2_.

#### 2.4.2. Baseline Disease Network

The baseline disease network represents the overall health trajectory of a population in terms of comorbidity progression over time. The admission histories of the selected two cohorts (such as *Cohort_T2DM&CVD_* and *Cohort_T2DM_*) are used to develop the baseline disease network. This study generated two such networks from two selected cohorts. The first disease network, referred to as *N_T2DM&CVD_*, is derived from *Cohort_T2DM&CVD_*; i.e., patients who are first diagnosed with T2DM and then diagnosed with CVD. The second disease network, referred to as *N_T2DM_*, is derived from *Cohort*_T2DM_; i.e., patients who are diagnosed with T2DM but not diagnosed with CVD. These two disease networks represent the health trajectories of the patients of *Cohort_T2DM&CVD_* and *Cohort_T2DM_*, respectively. Each disease network is generated by merging the individual disease networks of the patients of respective cohorts. In this way, the attributes of the node and edge of individual disease networks are summed up. The construction process of the baseline disease network is shown in Figure 1.

#### 2.4.3. Final Disease Network through Attribute Adjustment

After creating the two disease networks from two population cohorts, we merge them into one final network, referred to as the final disease network (*N_FD_*). This study applied the attribution theory to previously created baseline disease networks to generate the final disease network. Generally, attribution theory is the process of understanding the factors that are responsible for an event. These factors are used to predict the future occurrence of that event [49]. In this study, attribute adjustment gives weight to the comorbidities that have occurred significantly more in one cohort’s disease network compared to the other. The baseline disease networks *N_T2DM&CVD_* and *N_T2DM_* give us a scenario of attribution theory. If we find a disease that has a higher node frequency in *N_T2DM&CVD_*, this does not mean that this disease will be the risk factor for developing CVD in patients with T2DM, because that particular disease may have a higher prevalence in the T2DM patient cohort, *N_T2DM_*. However, instead of finding more prevalent comorbidities in *N_T2DM&CVD_*, this study focuses on looking for more prevalent comorbidities in *N_T2DM&CVD_* that are less prevalent in *N_T2DM_*. Thus, we look for more exclusive comorbidities or diseases in *N_T2DM&CVD_* compared to *N_T2DM_*, and in the process, we adjust for the attribution effect.

After applying attribution adjustment, the nodes and edges of the final disease network (*N_FD_*) are generated by merging the nodes and edges of *N_T2DM&CVD_* and *N_T2DM_*. The frequency of any node in *N_FD_* is calculated by finding its relative frequency increment in *N_T2DM&CVD_* compared to *N_T2DM_*. Similarly, the weight of edges is calculated. The final disease network (*N_FD_*) represents the health trajectory of the patients with CVD in T2DM patients, where the network attributes (node frequency and edge weight) show unique characteristics of progression towards CVD in T2DM patients in terms of comorbidities.

### 2.5. Procedure of the Proposed Framework

The input to the framework is the administrative data obtained from an Australian private healthcare fund. By applying the filtering criteria, this study selected two cohorts: *Cohort_T2DM&CVD_* and *Cohort_T2DM_*. The dataset included the ICD codes as the disease information of the patients. For *Cohort_T2DM&CVD_*, the admission histories for the patients with both T2DM and CVD are identified based on ICD codes. Then, all other ICD codes (related to the comorbidities from the Elixhauser index [16]) between the two diagnoses for a patient (when the patient was first diagnosed with T2DM and CVD, respectively) are considered to create the individual disease network. These individual disease networks are then aggregated to generate the baseline disease network: *N_T2DM&CVD_* for patients who were first diagnosed with T2DM and then diagnosed with CVD. Similarly, the baseline disease network, *N_T2DM_*, is created for patients who were diagnosed with T2DM but not diagnosed with CVD. Next, *N_T2DM&CVD_* and *N_T2DM_* are merged by applying attribute adjustment, and thus the final disease network (*N_FD_*) is created. Finally, graph theory and SNA are applied to *N_FD_* to understand the disease progression of CVD in patients with T2DM. The complete workflow of the proposed framework is illustrated in Figure 2.

## 3. Results and Analysis

After pre-processing and filtering the research dataset, we identified the T2DM and CVD patients by looking for the corresponding ICD codes (i.e., Table 2) in the admission history of the patients. A total of 4819 patients with T2DM-related ICD codes and 5731 patients with CVD-related ICD codes were found. Among the 4819 T2DM patients, we found 3908 patients who were diagnosed with T2DM but not diagnosed with CVD. On the other hand, among the 5731 CVD patients, we found 303 patients who were first diagnosed with T2DM and then were diagnosed with CVD at a later stage. Then, we found 172 patients out of 303 who have at least one ICD code related to comorbidities from the Elixhauser index [16]. Thus, a total of 172 patients were selected for the *Cohort_T2DM&CVD_*. Similarly, we found 822 patients out of 3908 who had at least one ICD code related to comorbidities from the Elixhauser index [16]. This study needs an equal number of patients for each cohort to create the final disease network; thus, a total of 172 patients out of 822 were randomly sampled for the *Cohort_T2DM_*, where the patients were diagnosed with T2DM but not diagnosed with CVD in their entire medical history. The flow of selecting the patients of cohorts is shown in Figure 3. The number of hospital admissions for *Cohort_T2DM_* (i.e., 1236) is slightly higher than the number of hospital admissions for *Cohort_T2DM&CVD_* (i.e., 1147).

### 3.1. List of Selected Comorbidities

The admission records of the patients were coded in both ICD-9-AM and ICD-10-AM as the dataset was based on an Australian healthcare context collected from a private health insurer in Australia. The data of the selected cohorts contained around 1000 different ICD codes. It was required to filter and group the ICD codes into comorbidities so that the nodes (i.e., ICD codes) of the disease network could be reduced into a small number of nodes. Therefore, an ICD mapping table for the comorbidity list was used to map each comorbidity with its relevant ICD codes. In this study, the Elixhauser comorbidity index was used to create the comorbidity list and the mapping table. The adapted Elixhauser comorbidity index [50] included 31 comorbidities. As the aim of this study is to understand the progression of CVD in patients with T2DM, we removed seven comorbidities (two for diabetes and five for CVD) related to T2DM and CVD. The translation table from the study of Quan et al. [46] was used to map of these 24 comorbidities to ICD codes. The ICD codes of the translation table were in ICD-9 and ICD-10 versions, which we manually tested with our dataset’s ICD-9-AM and ICD-10-AM versions. All of the ICD-9 and ICD-10 codes of the translation table matched with ICD-9-AM and ICD-10-AM, and hence no further modification was necessary. The list of selected comorbidities is given in Table 4.

### 3.2. Comorbidity Prevalence of N_T2DM&CVD_ and N_T2DM_

Using the above-mentioned lists of comorbidities, two baseline disease networks (*N_T2DM&CVD_* and *N_T2DM_*) were generated from two selected population cohorts which show the corresponding cohorts’ health trajectory. The node frequency of these baseline disease networks represents the prevalence of diseases.

Table 5 represents the most prevalent comorbidities from the two baseline disease networks. For the network *N_T2DM&CVD_*, the comorbidity conditions derived from the ICD codes were diagnosed any time after the diagnosis of T2DM but before the diagnosis of CVD, whereas the comorbidities for the network *N_T2DM_* were diagnosed before the diagnosis of T2DM. In Table 5, it is observed that the patients with both T2DM and CVD have a higher prevalence of comorbidities compared to the patients with the only T2DM. Additionally, there is a difference between the two baseline disease networks in terms of the most common comorbidities. The *N_T2DM&CVD_* baseline disease network shows a significantly high prevalence of renal failure, hypertension, chronic pulmonary disease, and obesity. These comorbidities or diseases are the risk factors for developing CVD within diabetic patients. These findings are consistent with the studies in the literature [7,51,52]. In this study, the frequency of nodes of the two baseline disease networks can give comparative insights about the prevalence of comorbidities between the patients with both T2DM and CVD and the patients with only T2DM. The average prevalence in N_T2DM&CVD_ is higher than the average prevalence in N_T2DM_, which shows that people with both T2DM and CVD have a greater health risk.

### 3.3. Attribution Effects on Final Disease Network

After generating the two baseline disease networks, this study generated the final disease network, *N_FD_*, through attribution adjustment. The final disease network shows the unique characteristics of CVD progression in patients with T2DM. *N_FD_* network assigned a higher score to comorbidities (i.e., node frequency) and their progression (i.e., edge weight) for those that are more prevalent in patients with both T2DM and CVD compared to the patients with T2DM only. In *N_FD_*, the node frequency and edge weight are normalized to the range of 0 to 1. Figure 4 represents the top 10 comorbidities with their normalized scores in terms of node frequency of *N_FD_*.

In the figure, the highest score of 1 for “renal failure” indicates that this disease was exclusive or more prevalent to the patients with both T2DM and CVD. The score for the comorbidities—i.e., fluid and electrolyte disorders, chronic pulmonary disease and hypertension—indicates that these diseases were the most prevalent in the *N_T2DM&CVD_* and had a small prevalence in the *N_T2DM_*. The other comorbidities, such as liver disease, solid tumor without metastasis and drug abuse, gained scores of less than 0.2. This refers to the small amount of differentiation for these comorbidity occurrences between the two baseline disease networks. Figure 4 suggests that the patients with T2DM have certain risk factors such as renal failure, chronic pulmonary disease, hypertension and fluid and electrolyte disorders that are associated with developing CVD. This observation is also consistent with the study of Nichols et al. [27] and Zhang et al. [33]. In the Australian context, this observation is also reported by the Australian Institute of Health and Welfare [53].

The top five pairs of the most prevalent disease progression of developing CVD in type 2 diabetic patients are shown in Table 6. The prevalent transitions of *N_FD_* show the comorbidities that are associated with the progress towards CVD in patients with T2DM. In Table 6, the highest transition (i.e., edge weight), which is from fluid and electrolyte disorders to renal failure, indicates two different problems of the body system. Fluid and electrolyte disorders are a deficiency or excess in key minerals (e.g., calcium and phosphorous) and electrolyte imbalances (e.g., sodium and potassium). The patients with T2DM develop a constellation of fluid and electrolyte disorders [54]. Also, potassium disorders can lead to the development of CVD [55]. On the other hand, renal failure is chronic kidney disease (CKD), and there is a strong comorbid relationship between CKD and CVD for a diabetic patient [33,56]. This reflects the fact that the transition between “fluid and electrolyte disorders” and “renal failure” may be a potential risk factor for the progression towards CVD in patients with T2DM. Additionally, the normalized weight of this transition is 1, which indicates that this is exclusive or more prevalent in patients with both T2DM and CVD compared to the patients with T2DM. The other top transitions in Table 6 are also more prevalent in patients with both T2DM and CVD. To test the statistical significance, we performed a Z-test on the weight of nodes and edges for the final disease network. The level of significance (*p*) was found to be less than 0.05 (*p* < 0.05).

The graph representation of the final disease network (*N_FD_*) is shown in Figure 5. To visualize and analyze the network, this study used the social network analysis software Gephi [57]. The nodes in the figure indicate the comorbidities or diseases. The size of the nodes and labels are proportional to the prevalence of the corresponding comorbidity. Renal failure, fluid and electrolyte disorders, chronic pulmonary disease and hypertension dominate the final disease network; this reflects the fact that these comorbidities can be risk factors of progressing towards CVD in patients with T2DM. The large number of edges in the network represents the transitions from one disease to another disease. The thickness of an edge is proportional to its weight.

### 3.4. Comparison of Network Measures for Three Disease Networks

This study performed a network comparison of the three disease networks. Several social network-based measures are calculated to understand the features of these networks, and the results are shown in Table 7. The table shows that the total number of nodes of the three disease networks are very close; this is because there are the same selective comorbidities for all three networks. Also, there is a high chance that each cohort has at least a few occurrences from these. The edge count in *N_T2DM&CVD_* is higher than the edge count in *N_T2DM_*, which shows that patients with both T2DM and CVD have relatively more admissions and more transitions between comorbidities in subsequent admissions. In addition, this may represent more exclusive comorbidities in subsequent admissions. The high graph density for the network of *N_T2DM&CVD_* also supports this suggestion. The graph density of *N_T2DM&CVD_* is higher than the graph density of *N_T2DM_*. This suggests that patients with both T2DM and CVD represent a higher admission burden and complex progression structure over subsequent admissions. The remaining two measures (e.g., average clustering coefficient and average path length) do not show important features in the present context.

## 4. Discussions

This study presented a network-based framework to understand chronic disease comorbidities. Administrative data are used in this study as they represent a unique source of information for patients’ medical conditions. Also, these databases are probably the best available source for understanding disease progression. Our study focused on the understanding of the progression of CVD in patients with T2DM. The final disease network generated from two baseline disease networks represents the health trajectory of the patients with CVD in T2DM patients, where the node frequency (i.e., Figure 4) and edge weight (i.e., Table 6) of the network show unique characteristics of progression towards CVD in patients with T2DM in terms of comorbidities. For instance, this study found some risk factors (e.g., renal failure, chronic pulmonary disease, hypertension and fluid and electrolyte disorders) that may be responsible for developing CVD in patients with T2DM [27,33,53]. Although these risk factors are well-known evidence for developing CVD among diabetic patients, this study provided further in-depth information about the transition between these comorbid conditions. For example, as illustrated in Table 6, we found that the transition between “fluid and electrolyte disorders” and “renal failure” has the highest weight, meaning that this transition may be a potential risk factor for the progression towards CVD in patients with T2DM. Apparently, no other studies in the present literature provide such in-depth information regarding the edge-level transition of comorbid disease conditions. The proposed framework for learning disease progression is flexible and can accommodate new sources of data for understanding the progression of multiple (more than two) chronic diseases.

### 4.1. Age and Sex Distribution of the Patients of Cohorts

Table 8 represents the age and sex distribution for both *Cohort_T2DM&CVD_* and *Cohort_T2DM_*. It is observed that the number of elderly patients (≥60 years) is higher than others in both cohorts. It is well known that the risk of developing T2DM and CVD in elderly patients is very high [58,59]. In Table 8, the number of female patients with both T2DM and CVD is higher than the number of female patients with only T2DM. On the other hand, the ratio of developing CVD in diabetic male patients is significantly lower. Regarding patients with T2DM, it has been shown that female patients have a higher risk of developing CVD than male patients [60]. In addition, the age for the first group (*Cohort_T2DM&CVD_*) is higher than the second group (*Cohort_T2DM_*) for each age range. This is obvious, as we considered T2D leading to CVD for the first group; on the other hand, the second group considered only T2D. Thus, the selected two cohorts could be uniform to analyze in term of age and sex. Additionally, male patients have a higher prevalence than female patients for both cohorts. This is consistent with the data released by the Australian Institute of Health and Welfare (AIHW) [53]. Thus, the research dataset is selected following the Australian government statistics.

### 4.2. Limitations of the Proposed Framework and Potential Future Works

This study has several limitations. The main limitation comes from the fact that the dataset used in this study uses real-world healthcare data. The quality of coding of these datasets is the main constraint because of the different coding criteria across different hospitals. The changing trend of policy is a cause of changes in the coding system. In addition, the expertise of clinical coders, funding and time constraints can affect the quality of coding. The data used in this study come from the hospital admissions and discharge summaries; thus, they do not include the GP (general practitioners) records and subsequent diagnoses. This may underestimate the comorbidity conditions of the patients. Additionally, the dataset collected from a private health insurer does not contain the information about the patients when they are admitted to a public hospital as public patients. However, most of these limitations are common for most administrative datasets. Another limitation of this study is that it does not include ischemic heart disease, acute myocardial infarction and stroke as a CVD since this study used the cardiovascular diseases mentioned in the Elixhauser comorbidity index. Also, this study cannot draw the time span between first and last hospitalization, as the selection criteria for cohorts limits the amount of data. Thus, the study considers the patients of cohorts with at least one hospitalization. Nevertheless, despite these limitations, the proposed framework could be useful for healthcare providers to obtain a better understanding of the progression of CVD in patients with T2DM.

As future work, the features extracted from the final disease network of chronic disease comorbidities can be utilized to develop a predictive model for future chronic disease. This can be implemented by comparing the final disease network with the individual disease network of a test patient. If the features of the test patient’s network match significantly with the features of the final disease network, the patient might be progressing on that chronic disease pathway.

## 5. Conclusions

This study presented a new framework to understand the comorbidity of two chronic diseases (T2DM leading to the development of CVD). For this, the proposed framework applied graph theory and social network analysis to administrative data based on the Australian context. Two cohorts (i.e., patients with both T2DM and CVD and patients with T2DM only) were selected to generate the corresponding baseline disease network. The final disease network was generated from these two baseline disease networks through attribute adjustment. This can represent the overall health trajectory of the patients of both cohorts in terms of comorbidity progression over time. This was then analyzed by network measures. As a result, this study found some risk factors (i.e., renal failure, fluid and electrolyte disorders, chronic pulmonary disease, hypertension and obesity) that could be associated with the development of CVD in patients with T2DM. The network-based methods demonstrate the effectiveness of the proposed framework for understanding the progression of CVD in patients with T2DM. Also, this framework can be tested for other groups of comorbidities to understand their progression. Thus, the study can help healthcare providers to understand high-risk diseases and the progression patterns between the recurrence of T2DM and CVD. Also, it can help providers to manage healthcare resources efficiently.

## Figures and Tables

**Figure 1 ijerph-17-00596-f001:**
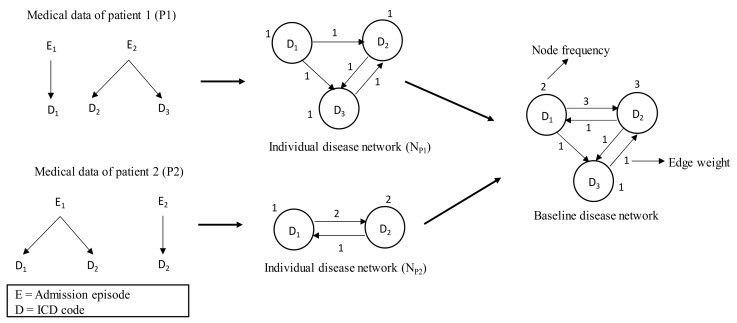
Construction of baseline disease network. First, individual disease networks are developed from medical data of the corresponding patients and are then aggregated to generate the baseline disease network.

**Figure 2 ijerph-17-00596-f002:**
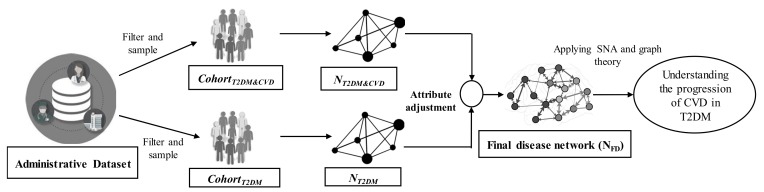
Proposed framework to understand the progression of cardiovascular disease in patients with type 2 diabetes. SNA: social network analysis.

**Figure 3 ijerph-17-00596-f003:**
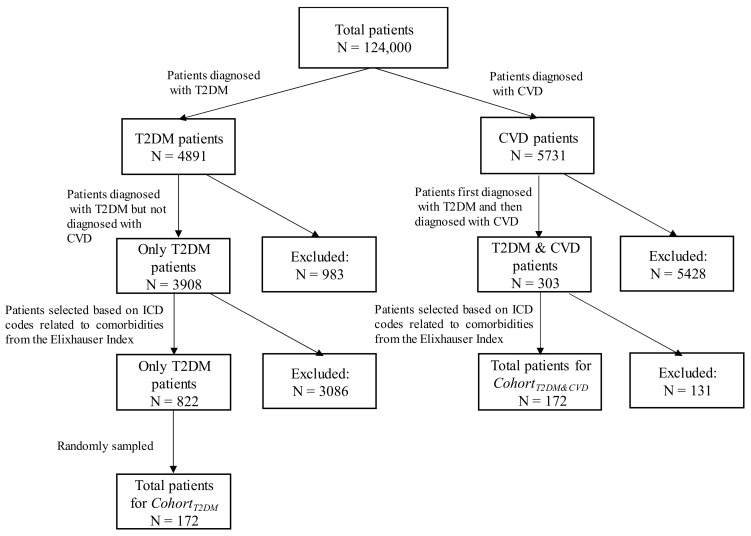
Flow diagram of selecting the patients of cohorts.

**Figure 4 ijerph-17-00596-f004:**
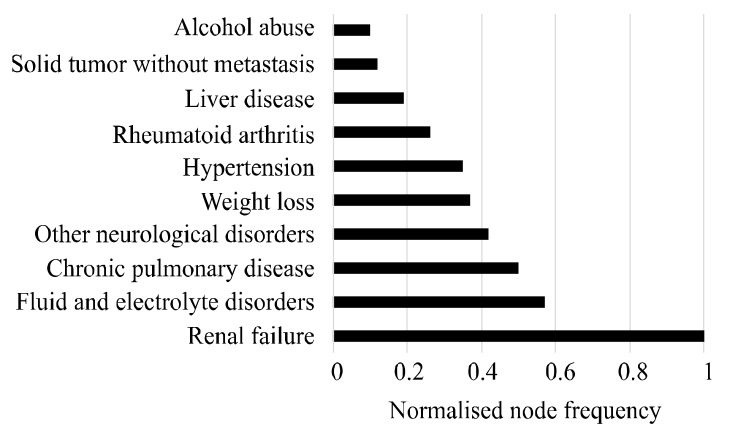
Top 10 comorbidities that attributed most to the progress of CVD in patients with T2DM.

**Figure 5 ijerph-17-00596-f005:**
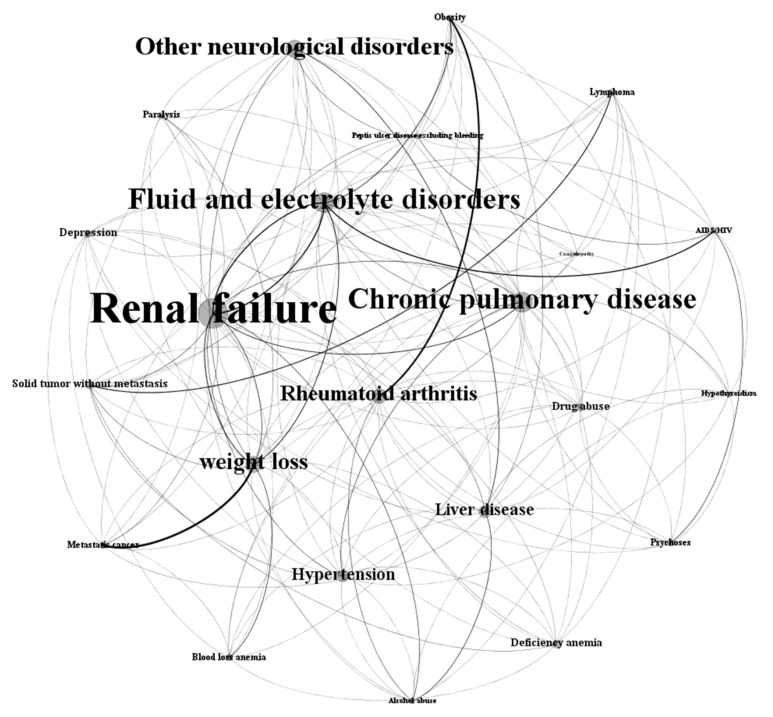
Final disease network of T2DM patients progressing towards CVD. The node size and labels are proportional to the prevalence of the corresponding comorbidity. The thickness of an edge between two comorbidities is proportional to its weight.

**Table 1 ijerph-17-00596-t001:** Contents of the administrative data used in this study. ICD: International Classification of Diseases.

Dataset Contents	
Patient ID	Claim ID
Gender	Episode ID
Age	Diagnosis procedure code
Location postcode	ICD types and codes
Provider ID	Diagnosis-related group (DRG) codes
Admission and discharge date	

**Table 2 ijerph-17-00596-t002:** ICD-9-AM and ICD-10-AM codes for cardiovascular disease (CVD) and type 2 diabetes mellitus (T2DM) (Adapted from Quan et al. [46]).

Comorbidity	ICD-9-AM Codes	ICD-10-AM Codes
Congestive heart failure	398.91, 402.11, 402.91, 404.11, 404.13, 404.91, 404.93, 428.x	I09.9, I1.0, I13.0, I13.2, I25.5, I42.0, I42.5–I42.9, 143.x, 150.x, P29.0
Cardiac arrhythmias	426.10, 426.11, 426.13, 426.2–426.53, 426.6–426.28, 427.0, 427.2427.31, 427.60, 427.9, 785.0, V45.0, V53.3	I44.1–I44.3, I45.6, I45.9, I47.x, R00.0, Roo.1, R00.8, T82.1, Z45.0, Z95.0
Valvular disease	093.2, 394.0–397.1, 424.0–424.91, 746.3–746.6, V42.2, V43.3	A52.0, I05.x–108.x, I09.1, I09.8, I34.x–I39.x, Q23.0–Q23.3, Z95.2–Z95.4
Pulmonary circulation disorders	416.x, 417.9	I26.x, I27.x, I28.0, I28.8, I28.9
Peripheral vascular disorders	440.x, 441.2, 441.4, 441.7, 441.9, 443.1–443.9, 447.1, 557.1, 557.9, V43.4	I70.x, I71.x, I73.1, I73.8, I73.9, I77.1, I79.0, I79.2, K55.1, K55.8,K55.9, Z95.8, Z95.9
Type 2 diabetes mellitus	250.0–250.3, 250.4–250.7, 250.9	E11.0, E11.1, E11.2–E11.9

**Table 3 ijerph-17-00596-t003:** Selection criteria for both cohorts.

Selection Criteria for *Cohort_T2DM&CVD_*	Selection Criteria for *Cohort_T2DM_*
-Must be first diagnosed with T2DM and then diagnosed with CVD.	-Must be diagnosed with T2DM but not be diagnosed with CVD.
-Must have at least one or more admissions after the date of first diagnosis with T2DM but before the date of diagnosis with CVD.	-Must have at least one or more admissions before the date of first diagnosis with T2DM.
-For each admission, must have at least one or more ICD codes related to comorbidities from the Elixhauser Index.	-For each admission, must have at least one or more ICD codes related to comorbidities from the Elixhauser Index.

**Table 4 ijerph-17-00596-t004:** Elixhauser comorbidity list used in this proposed framework.

Comorbidities	
Hypertension, uncomplicated	Solid tumor without metastasis
Hypertension, complicated	Rheumatoid arthritis/collagen vascular diseases
Paralysis	Coagulopathy
Other neurological disorders	Obesity
Chronic pulmonary disease	Weight loss
Hypothyroidism	Fluid and electrolyte disorders
Renal failure	Blood loss anemia
Liver disease	Deficiency anemia
Peptic ulcer disease excluding bleeding	Alcohol abuse
AIDS/HIV	Drug abuse
Lymphoma	Psychoses
Metastatic cancer	Depression

**Table 5 ijerph-17-00596-t005:** Top 10 most prevalent comorbidities for patients with both T2DM and CVD, and patients with only T2DM. The prevalence refers to the number of admissions that have ICD codes related to those comorbidities.

Comorbidities for *N_T2DM&CVD_*	Prevalence	Comorbidities for *N_T2DM_*	Prevalence
Renal failure	430	Depression	331
Solid tumor without metastasis	300	Metastatic cancer	265
Hypertension	102	Solid tumor without metastasis	205
Peptic ulcer disease excluding bleeding	71	Obesity	114
Fluid and electrolyte disorders	63	Peptic ulcer disease excluding bleeding	40
Other neurological disorders	60	Drug abuse	30
Chronic pulmonary disease	41	Paralysis	22
Liver disease	24	Psychoses	18
Obesity	21	Hypertension	12
Weight loss	17	Other neurological disorders	09

**Table 6 ijerph-17-00596-t006:** Top five most prevalent transitions between comorbidities in the final disease network.

Initial Condition	Next Condition	Normalized Weight
Fluid and electrolyte disorders	Renal failure	1
Weight loss	Fluid and electrolyte disorders	0.80
Renal failure	Chronic pulmonary disease	0.70
Other neurological disorders	Liver disease	0.65
Renal failure	Weight loss	0.61

**Table 7 ijerph-17-00596-t007:** Different network measures of three disease networks.

Network Measures	*N_T2DM_*	*N_T2DM&CVD_*	*N_FD_*
Number of nodes	22	21	23
Number of edges	80	120	166
Graph density	0.20	0.30	0.22
Network diameter	4	4	4
Average clustering co-efficient	0.49	0.63	0.55
Average path length	2.11	1.90	1.91

**Table 8 ijerph-17-00596-t008:** Age and sex distribution of the population for both *Cohort_T2DM&CVD_* and *Cohort_T2DM_*.

	*Cohort_T2DM&CVD_* Population %	*Cohort_T2DM_* Population %
Age		
0–30	0	0
31–40	0.58	0
41–50	1.16	0.58
51–60	4.65	16.86
61–70	18.60	25
71–80	31.39	26.74
81–90	35.47	24.42
91–100	7.56	5.23
≥101	0.58	1.16
**Gender**		
Male	59.30	69.65
Female	40.70	30.35

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
