# Peer review of "A Framework to Understand the Progression of Cardiovascular Disease for Type 2 Diabetes Mellitus Patients Using a Network Approach"

_ijerph, 2020, doi:10.3390/ijerph17020596_

Round 1

Reviewer 1 Report

The manuscript has been improved substantially. I have no further comments.

Reviewer 2 Report

The manuscript has been revised according to the reviewers' comments.

This manuscript is a resubmission of an earlier submission. The following is a list of the peer review reports and author responses from that submission.

Round 1

Reviewer 1 Report

The authors conducted retrospective longitudinal study examined comorbidity prevalence among patients with both T2DM and CVD and patients with only T2DM. The prevalence of renal failure, fluid and electrolyte disorders, hypertension and obesity was significantly higher in patients with both CVD and T2DM than patients with the only T2DM. These analyses were performed using graph theory and social network analysis techniques. Analytical technique is unique but the result the authors showed in this manuscript that CKD, volume over, hypertension and obesity are the risk for CVD among diabetic patients is already well-known evidence.

There are a couple of issues that should be addressed.

Major Concerns

Please emphasize and clarify what is new and different from usual risk factor analyses in this study/analyses. Database is large (total 1,23,983 patients) but the authors only use 172 patients from each cohort (total 344 patients). This reviewer does not think this is the best way to analyze the risk factors for CVD among diabetic patients.

The authors should explain how to manage patients who hospitalized >3 times in methods.

In results section; p7 line275, authors mentioned “A total number of 3,908 patients with T2DM”. There are 822 patients in CohortT2DM&CVD and 281 patients in CohortT2DM that means rest of 2,805 patients (72%) were diagnosed with T2DM and CVD simultaneously at first administration? This reviewer thinks this proportion does not represent real world. The authors should explain the patient inclusion and exclusion criteria in detail using fish bone chart, and discuss about the distorted distribution between T2DM and CVD among hospitalized patients

Why “172” patients were selected from CohortT2DM&CVD; 822 patients and from CohortT2DM; 281 patients. The author explained how to decide the number of patients.

Ischemic heart disease, acute myocardial infarction and stroke including brain hemorrhage and brain infarction are also the main cause of CV events. Why did the authors exclude these diseases?

Please explain the meaning of number in table 5. There are “331” depression among 281 patients from CohortT2DM. This reviewer cannot understand this table.

The author should show the difference of baseline characteristics between two cohorts including age, time span between first and last hospitalizations, number of hospitalizations and etc. and discuss how these factors can affect the results in this study and whether these two cohorts would be uniform cohorts to comparable/analyze.

Minor concerns

The author could separate the results section and the discussion section that will be easy to read.

Reviewer 2 Report

In this study authors used graph theory and social network approach to examine the progression of cardiovascular diseases in patients with type 2 diabetes. The study cohorts that is patients with T2DM without cardiovascular diseases and with both T2DM and cvd were selected from administrative dataset (n=172 patients in each cohort). The results demonstrate that the prevalence of comorbidities such as renal failure, hypertension, chronic pulmonary disease, and obesity is higher in patients with T2DM who subsequently develop cvd than in those who do not develop cvd. The most prevalent pair of comorbidities in patients with T2DM who subsequently developed cvd were fluid electrolyte disorders followed by renal failure.

The topic is of interest. The research approach was original and fruitful. The manuscript is overall well-written. There are some minor corrections to be made.

Lines 71-72: the first sentence of the paragraph is confusing; please correct. Line 80 “severity of the patient” should be revised to “severity of the disease”. Line 82: “raking” should be corrected to “ranking”.